# Reskilled and Integrated, but How? Navigating Trauma and Temporary Hardships

**DOI:** 10.3390/ijerph192013675

**Published:** 2022-10-21

**Authors:** Cihan Aydiner, Erin L. Rider

**Affiliations:** 1Homeland Security Program, Department of Security and Emergency Services, College of Arts & Sciences, Worldwide Campus, Embry-Riddle Aeronautical University, Daytona Beach, FL 32114, USA; 2Department of Sociology and Political Science, College of Social and Behavioral Sciences, Jacksonville State University, Jacksonville, AL 36265, USA

**Keywords:** forced migration, exile, well-being, highly educated migrants

## Abstract

Immigrants are often pressed to show how they will contribute to a host country, thus proving through their conditions of entry and human capital whether they will be perceived as an asset or burden, and this is juxtaposed with the host country’s institutions offering an improved quality of life, mainly through employment. Seeking employment is often a key factor to be economically assimilated, and in the case of highly educated Turkish migrants, the opportunity to reclaim their previous professional and quality of life statuses. Based on qualitative research, we have examined the experiences of highly educated Turkish people (n = 42) in the recently forced migrant population. Following events including terrorism and the coup on 15 July 2016, Türkiye experienced the highest forced migration in her history. With exiled Turkish migrants, the forced aspect of their migration prompts them to seek a host country that provides safety, and they are also driven to transfer their educational degrees and professional credentials. However, changing careers to become educated and certified in new fields takes time and resources, contributing to a fluctuating economic status and loss of well-being. Once this is regained, their economic situation is improved, but there is still the lost time from the immigration and transfer period. Thus, the process has positive and negative components, but understanding this nuanced process provides opportunities for policy reform that can shorten the time of re-education, increase employability, and support well-being.

## 1. Introduction

### 1.1. Migration of Large Turkish Population to the USA

Immigrants, especially forced ones, generally prefer neighboring countries to migrate to due to main reasons such as affordability, proximity, and similarity to their origin states. Moving too far away from the origin country requires more economic, social, and cultural capital. In the case of modern Türkiye, large groups of Turkish migrants have taken residence within Europe mainly for economic reasons and family reunification in the 1960s. Turks contribute to the highest immigrant population, with more than five million in Europe now [1].

Due to immigration laws, Turkish immigrants to the USA were a smaller migration population, but were more educated, with around 23,000 in the same period [2]. Following World War II, the U.S. attracted many professionals with its rise in science and technology. Highly skilled and educated Turks, including doctors, engineers, and academics, also took their place in this brain drain in the USA until the 1980s and increased the population up to 50,000 [3,4]. Between the 1980s and 2010, the Turkish population reached close to 200,000 based on US census data (2010). During this period, Türkiye faced a military coup and the Kurdish conflict. As a result, Kurdish-origin Turks and other less educated and semi-skilled Turks migrated to the U.S. due to political unrest and economic reasons [2].

The Turkish migration to the U.S. following the coup in Türkiye on 15 July 2016 was dramatically different from previously forced migrations. The foundation of democracy has been undergoing recent threats, and authoritarianism is rising in many countries. Some semi-democratic regimes are using staged events such as coups or terrorism for the sake of their survival, transition to autocracy, and empowerment. Türkiye is an explicit example of this transition process under Erdogan’s rule. On 15 July 2016, staged events such as the coups were organized and directed by Erdogan’s government and its partners to oppress opposing groups and people on the side of democracy, freedom of speech, and rule of law. Erdogan described the events of 15 July as a “gift from God”. Regarding the critical events and indications of democratic erosion in Türkiye before and after 15 July 2016, coup-like events provide a larger picture of the causes of targeting some people in the country. Those opposing groups on the side of democracy have been silenced, isolated, jailed, exiled, and/or forced to migrate systematically to accomplish the transition process to the anti-democratic regime by Erdogan’s government. Specifically, the ruling AKP (Adalet ve Kalkinma Partisi—Justice and Development Party) declared a state of emergency, imprisoned or fired hundreds of thousands of high-profile civil servants, and closed thousands of schools and hundreds of news outlets. During the crackdown, many officials in Western countries, including diplomats, foreign ministry officials, and academics, refused to return home following government orders due to a lack of evidence and the risk of imprisonment. Some of them returned and experienced imprisonment or mobbing, abuse, and eventual dismissal from their posts. This political climate has prompted former state officials to consider the option of permanent residence in the host country. As a result of these political conditions, Türkiye experienced the third highest immigration pattern in her modern history. Over 250,000 Turks emigrated in 2017 after the events, which was a 42% increase from 2016. Asylum applications worldwide increased by 10,000 to over 33,000 in 2017 [5]. The number of Turkish immigrants obtaining lawful permanent resident status in the USA was approximately 5000 yearly from 2017 to 2021. However, the number of individuals granted asylum affirmatively was only 15 in 2017. Asylum rates increased to 475 in 2018, 1714 in 2019, and 1568 in 2020 [6]. The Turkish population is estimated to be around 350,000 in the U.S. currently, including increased unauthorized arrivals, especially through the El Paso/Texas entryway, reaching more than 10,000 Turkish nationals in 2022 [7].

The main reason for professionals’ migration was opposition to the ruling party. Almost all of them were highly educated, skilled, and legal migrants. Professionals included judges, teachers, doctors, journalists, police officers, ministry officials, and military career officers, including cadets and retired officers. If they were not arrested, most of them were denied passports, disqualified, humiliated, and stamped in the country. This atmosphere has forced many to leave the country through legal or illegal means to Europe and North America. Many of them were fluent in English and familiar with life in the U.S. due to having educational and occupational experience prior to the 2016 events. However, despite some familiarity with the U.S., most of them had never envisioned living in another country because their identity and loyalty were centered on serving their homeland (Aydiner and Rider 2021). Many of these displaced professionals have been granted special visas, green cards, asylum, or refugee status. Managing the migration process and providing suitable jobs for this talented immigrant group is, therefore, a challenge [8].

### 1.2. Purpose of the Research

The recently immigrated Turkish migrant population creates a compelling case for researching higher education connections and the administration of justice in migration policies/practices related to highly educated individuals’ integration into the U.S. as well as supporting their well-being. Understanding this nuanced process provides opportunities for policy reforms that can reduce re-education and recredential time, increase employability, and promote well-being. We anticipate there are benefits to the U.S. in employing this professional labor group and inclusion strategies for new immigration populations from different countries following global crises and events similar to those experienced by exiled Turkish professionals. Uniquely, this professional labor group occupies a higher economic status as a group of professional, skilled workers with presumably greater economic options in the U.S. and a strong likelihood of permanent integration.

The most important research questions guiding this study were: What are the long-term effects of immigrants’ new career paths, their strategies/solutions to cope with policy issues, and the psychological issues that follow them? and What are the outcomes for the temporary deskilling of highly educated immigrants? The current study focuses on Turkish emigration following the 2016 coup-like event. This migrant population is unique, given that they were forced to flee their relatively stable lives built around their family, education, and profession in order to seek safety. Their forced migration was unplanned and required significant maneuvering to re-establish their lives in a foreign country. In acknowledgment of the dire factors of the immigration process, many Turkish immigrants had high human capital with regard to education and their profession. Higher human capital tends to indicate the self-selection of a host country that can offer adequate labor opportunities. However, for Turkish immigrants, despite their high levels of education and professional status, the forced aspect of their migration and the level of urgency complicated their immigration experience and ability to secure stability, maintain their high human capital, and regain their former professional status. This study examines the economic assimilation experiences of forced yet highly educated Turkish immigrants to better understand complex factors that either aid or hinder economic integration in a destination society. This project suggests direct benefits for the management of highly educated migrants by learning the characteristics of the migrant population and proposing solutions/innovative alternatives for inclusion and integration. The study helps to understand the development of a policy reform by showing the nuanced differences of the population. This may support the creation of a fast-tracked integration model by collaborating with employers and organizations to strategize efforts for streamlined accreditation and training processes.

## 2. Literature Review

Immigration studies have largely focused on the dichotomy between push and pull factors that impact immigrants in different ways based on their immigration type—voluntary and involuntary [9,10]. It has long been supported that economic migrants (voluntary) have greater decision-making ability and financial capital planning for migration [10,11]. For political refugees, their migration journey is determined often immediately with little opportunity to plan [12,13,14]. The unique circumstances of Turkish migrants can be predicted based on push and pull factors. Push factors are related to their country’s political and economic instability and lack of safety, while pull factors are associated with safety and economic opportunity in the host society. Regarding social capital, previous experience in the U.S. and other Western countries, educational and professional skill sets, degree of English fluency, and social networking with other Turks, these factors can afford Turkish immigrants a smoother integration transition [8,15]. However, some factors could be barriers; these include challenges associated with forced migration and trauma, foreign credentials, lack of work experience in a host country, and less direct professional and social networks with native workers.

The literature has characterized the challenges of immigrant assimilation in a variety of contexts, such as social, professional, and civic, and the nature of the immigration process initially, either forced or voluntary. In the literature, it is often presumed that forced or voluntary migrants with little human capital will face greater barriers to improving their quality of life and social and economic integration [16]. On the other hand, destination countries tend to advertise for a need to accept highly educated professionals to fill gaps in work industries [17,18,19]. Sapeha [20] finds “during the application process, the Canadian immigration system allocates more points to immigrants with higher levels of education. This assumes that those with higher human capital would have a smoother process of integration into the receiving society. However, findings from the present study do not support this assertion”. This study finds levels of satisfaction to be lower [20]. In the case of highly educated Turkish immigrants, the assumption of being sought after for their educational background, skill sets, and professional experience underscores another factor that their economic integration is not always easy nor streamlined. Specifically, Western developed countries’ demand for professional skilled immigrants assumes there will be less barriers and public welfare concerns [21], enabling immigrants to efficiently integrate into the skilled labor market, which is an ultimate mutual benefit to destination countries’ skilled employment and immigrants’ quality of life. However, several studies have come to find that foreign credentials hold lower value and are not comparable to native citizens’ credentials [22,23,24]. Significantly, Sweetman, McDonald, and Hawthorne find that this leads to consequences for immigrants, specifically “lower earnings and job dissatisfaction” and “to the host country in terms of lost productivity” [25].

There are several factors unique to highly educated/skilled migrants that pose barriers to economic integration and workplace participation: the foreign credential recognition process [25,26], occupational mismatches [27,28], hiring chances [26], and long-term earnings gaps [25,29]. These barriers have been generally attributed to or categorized as ethnic penalties [30,31] and are noticeable in comparison to native citizens. The foreign credential recognition process is complex as it is not standardized across professions. In general, foreign degrees are often more challenging to verify regarding equivalency and quality in comparison to the ease of determination for native citizen degrees. Research on STEM-educated immigrants by Boyd and Tian finds that in comparison to Canadian-educated citizens, immigrants educated in “Canada, the U.S., the U.K., and France, do earn less, but the earnings gap are lower than for immigrants educated elsewhere” [32]. As Boyd and Tian assert, “degrees from Eastern Europe and Asia are not as portable” [32]. These findings reveal both ethnic penalties and assumptions about which countries’ degrees are viewed as equivalent. Here, Western countries standardize foreign degrees from other Western-identified countries. Michalikova, in her research on Eastern European immigrants, finds that they are hindered by socioeconomic differences based on their educational systems, pre-emigration labor market experiences, and the language barrier [31].

Certain professions, such as those in the medical field, encounter a more arduous process that may require education and training to be done in the destination society rather than be transferred in, while other professions that do not have licensure may find an easier transfer process. Occupations with associated credentialing tend to be more difficult for immigrants to access, and immigrants are less likely to gain employment in these professions [33]. Despite these access barriers, these jobs tend to be coveted because they are associated with “higher wages and their wage disadvantage is lower” [33]. This leads to the previously mentioned notion that host countries can attract highly educated immigrants with the promise of skilled professions, but in reality, the access to these jobs and transferability are low. The time and money invested in an occupation, thus, may not be matched with opportunities to continue in that occupation in the destination society. Notably, in a study examining the outcomes of occupational licensure, Redbird and Escamilla-Garcia found a positive linkage in the ability for licensure to “increase inclusion and transferability of educational credentials” [34]. Other research has shown similar findings that licensure can enable immigrants to access professions based on verification (see Fu and Hickey [35]). Similarly, Haberfield et al., in their research on highly skilled German immigrants, found that factors such as highly skilled, transferability of human capital, self-selection of the host country, and reception in the host country all added to their ability to reach “full earnings parity with natives of similar attributes” [36]. Other research has added that immigrants with “standardized and occupation-specific profiles” can access greater advantages [26].

Immigrants may also find themselves unable to continue in their profession due to barriers in the credential recognition process, lack of available jobs in the destination society, and time spent waiting for re-education/certification. Some immigrants may choose to seek out a new occupation, thus embarking on new educational and skill development. These circumstances demonstrate problems associated with an occupational mismatch [28,37]. In this sense, immigrants’ education, skills, and professional experience are unused and potentially wasted. Guo observes a glass gate effect in which “professional communities” restrict qualified immigrants because of the perception that their education and skill credentials are deemed lower [23]. This is further compounded by the glass door and glass ceiling effects; former education is not perceived as transferrable [23]. In a study by Dahlstedt, they found differences in the mismatch between vocationally educated and generally educated populations. Specifically, “the vocationally-educated population are more likely to be occupationally matched, but also that they are unlikely to advance past their educational level”, while the reverse is true for the generally educated population who experience “occupational mismatch and advance to employment exceeding their educational level” [27]. Some of these initial barriers can be improved with recertification or by seeking new education or credentials and with built-up experience in the workforce based on settlement length [38,39]. Further, there may be other emerging barriers related to soft skills, such as degree of cultural awareness, language barriers [40], and cultural factors.

The inability to have foreign credentials transferred, thus leading to short-term or long-term differences in jobs or careers, can lead to lower levels of occupational commitment. Changing career pathways may indicate that the second career choice is less complementary to an immigrant’s preferred career and professional identity and instrumental to their need to seek a career for economic capital. Referring back to occupational mismatch, the reason for changing careers is for economic opportunity, not one’s connection to a particular field. For example, in our previous research [8] on a separate sample of Turkish immigrants, individuals made new career selections based on the current demands of employers, thus seeking out industries in higher demand, careers that would have a shortened timeframe between training to job placement, and jobs that pay well. The instrumental reason to seek a career that brings about economic stability contradicts one’s passion for the field. It may be that the original career was closer to one’s interest and the secondary career was instrumentally based. Although, it could very well be that the first career was also instrumentally based. Research has shown that, over time, lower levels of occupational commitment led individuals to end career positions prior to retirement [41].

In recognizing that highly educated immigrants seek out new career pathways in order to speed up their economic integration into a destination society, we can compare the amount of time invested in pursuing two or more career fields. Previously, Turkish migrants were educated and had professional employment in Türkiye, but following their forced immigration experience, for some, they sought new careers with re-education or training that could take one or more years to complete. During this time, they were typically employed in the secondary sector until establishing their new career. Although they found temporary employment, wages were low, work conditions were likely to be more exploitative and demoralizing, and difficulties emerged with balancing work–education–family. Over time, this temporary hiatus from their career demonstrates an overall earnings gap impacting both earned income and retirement benefits.

This study extends the literature on economic integration by examining the U.S. context specifically and migrants from different career types. It also broadens the focus to include the pre-conditions for migration, and a career focus related to the initial adjustment to the U.S. and the degree of success in regaining professional status. Similar to other studies, the barriers, challenges, and any negative outcomes are understood through a focus on immigrants’ agency.

## 3. Materials and Methods

Much of the literature tends to focus on the forced aspect of migration experienced by undocumented migrants or refugees with lower social capital levels; however, this study will examine the unique conditions of highly educated, forced migrants that offer the U.S. a professional labor stock, and how they navigate the immigration process to regain status and reach economic integration. Instead of focusing on the time of immigration to the destination country, we have broadened our focus to the holistic migration journey of the immigrants, starting with the details of societal and individual events forcing them to move to the decision-making process and, finally, to the economic and psychological process of regaining their status. Based on recent qualitative data, including tracking some interviewees from our previous study in 2019, we have interviewed 42 (12 of them tracked from our first study) highly educated Turkish people in the U.S. In our first comparative study, we interviewed 30 forced or exiled Turkish people in eight different countries, including six European countries, Canada, and the U.S. Half of the respondents were from the U.S. [8]. They were living in historically highly Turkish-populated areas (more than 10,000) in the U.S. in the order of: New Jersey/NY, Dallas/TX, Houston/TX, California, and Virginia. The Turkish population increased between 15% to 50% from 2015 to 2020 in these selected cities [6]. These longitudinal data efforts have increased our ability to both generalize findings and garner rich information. The focus of our study is specifically on highly educated Turkish people who moved or were exiled to the U.S. due to increasing oppression, economic instability, unhappiness, and polarized relations after the 15 July 2016 events in Türkiye. Since we worked in-depth with vulnerable individuals who were targeted in their homeland, our methodology was shaped by the sensitive political and psychological circumstances of the forced and/or exiled community. The study is qualitative.

### 3.1. Recruitment Procedures

The first author is a Türkiye-born researcher who came to the USA one year before the coup-like staged events and has a good network with the recent Turkish exiled community. The study and its procedures have been approved by the Institutional Review Board (IRB). The collected data are from four main areas in the U.S., consistent with the most Turkish-populated regions. These cities are New Jersey, Houston, Dallas, and Virginia Beach. The researchers created a flyer, an informed consent form, and a detailed call about the study to recruit interviewees and provided Amazon gift cards as incentives. The requirements to participate in the study are stated as follows: being a highly educated Turkish person who has a college/university degree from a 4-year university/college or attends a college/university currently and who moved, decided to stay, or is planning to stay in the U.S. within the last six years. The researchers disseminated flyers to Turkish network groups and organizations. Some organizations, including the Blue Tulip Human Rights Center, based in Virginia and founded by Turkish veterans and American professionals, featured the study and shared it with their networks. This organization describes its mission as “increasing local and global awareness of sufferings of isolated and silenced women and children” on its webpage [42]. They have scholarships, psychological support, and empowering women and human rights awareness programs, especially for Turkish migrants who moved to the U.S. following the 15 July events. As a result, many of the volunteers in their network decided to support the study. The lead researcher also has a connection with Turkish Professionals US, which is very active in social media and has almost 9000 members. They support each other in information sharing, job searches, and integration into the U.S. Some of the interviewees were affiliated with this group. Lastly, the researcher also has connections with former state officials (i.e., bureaucrats and senior NATO officers). In accordance with our snowball sampling techniques, they also supported the study, and some shared their experiences during exile and the forced migration process.

Interviewees signed their informed consent forms before the face-to-face interviews. Participants were assured of the confidentiality of the procedure, including the fact that no personal identifiers were included in the transcripts, each interviewee was assigned a pseudonym, and results were reported as grouped data. Additionally, participants were informed they would have access to the study results, and Amazon gift cards were shared after the interviews. All interested participants gave informed consent and completed their interviews.

### 3.2. Interviews

Semi-structured interviews were conducted between May and September 2022. Forty-two people agreed to and finished an interview. Twelve of the interviewees were tracked among fifteen individuals in the U.S. from our previous comparative study conducted in 2019. Three of them were not available in our travels to target regions [8].

We conducted audio interviews in places based on participants’ requests (i.e., participants’ offices and library rooms) to protect their privacy. Furthermore, we conducted video interviews with the eleven volunteered participants to better observe their body language and facial expressions in places based on participants’ requests. The risks of participating in this study were no more than what was experienced in daily life. The participants of the interviews did not have to answer any questions they were uncomfortable answering. Interview sessions, on average, took approximately one hour. However, some of the interview sessions lasted up to two hours due to the speaking pace of the participant, some emergency calls during the recording, or individuals having a hard time responding to emotional/sensitive questions. Although all interviewees were fluent in English, the interview language was Turkish to completely reflect the psychological and emotional circumstances of the participants.

The semi-structured interview guide enabled the interview to flow in a conversational way, and interviewees were asked to share their stories and experiences about the process. However, if the interviewees did not elaborate or had a limited response, we provided follow-up prompts such as, “How did you earn money until getting your work permit?” or “Have you ever had PTSD in your migration process?”

We observed body and facial expressions during the sessions and analyzed them along with the transcripts. Qualitative data were transcribed, coded, and analyzed using Atlas Ti 8 following the grounded theoretical design of Charmaz [43]. The coding procedure involved highlighting phrases, terms, and topics within the transcribed text that were repeated during unstructured conversation and grouping them under themes.

### 3.3. Demographics

Forty-two highly educated people participated in the study. They live in New Jersey (11), Dallas (9), Houston (8), Virginia (7), and 7 of them live in other places, namely California, Pennsylvania, Chicago, and Florida. Some of them, including tracked individuals from the previous study, had moved to new places different than their first location in the U.S. due to better jobs or living opportunities. They were all legally in the U.S.

The participants’ ages were in the thirties to mid-fifties. Ten of them were 30–35 years old, seven of them 35–40 years old, thirteen of them 40–45 years old, five of them 45–50 years old, and seven of them 50–55 years old. Most participants were male (37), and five of them were female. All of them were married with Turkish spouses because having a good marriage was one of the social criteria selected for representing roles abroad for highly educated Turkish exiles. Thus, we also asked questions about their spouses’ experiences. During the July 15 events, 22 of the participants were in Western countries due to professional roles representing Türkiye, pursuing a graduate degree, or on short-term vacation following job-related meetings/courses. All of them were called back within one year following the events in Türkiye with state orders to return immediately. Three of them returned and worked under pressure and in extraordinary situations for a while. Then, they understood that they did not have any alternatives to being imprisoned and, thus, decided to return to the U.S. Other participants had enough time to evaluate what could happen to themselves if they returned. Because those who return immediately have faced jail, probation, torture, mobbing, or abuse for flimsy reasons, they decided to stay in their locations until they saw a clear picture of the events. However, the state of emergency was extended seven times for three months and ended after two years. During this period, the government started to dismiss (including our interviewees) hundreds of thousands of state officials in critical positions (e.g., president and members of the supreme court, high-level judges, lawyers, opposition politicians/leaders, military generals, heads of police departments, human rights defenders, and journalists). Following these events, Türkiye transitioned from a parliamentary system to a Turkish-type presidency system, and Erdogan became the first president. The highly educated participants who were abroad understood that they were in exile until democracy came back to their homeland. Four out of twenty-two participants who were abroad moved from European countries to the U.S. due to their positive feelings during their previous experiences in the U.S., specifically including the language barrier in Europe, job opportunities, and a better future for their children. Twenty participants were in Türkiye during the events. Most of them have experienced difficult times, including imprisonment, abuse, mobbing, and social death, and experienced many challenges during their journey to the U.S. The jobs of the interviewees were high-level military officers, physicians, engineers, academics, and college students in closed institutions during the 15 July events.

## 4. Results and Discussion

### 4.1. Perspectives and Experiences of Highly Educated Exiled Turks in Their Journey to the USA

We examined the journey of Turkish professionals under three themes that emerged from the interviews: issues migrating with them, short-term poverty, and filling the gap.

### 4.2. Issues Migrate with Them

This section focuses on Turkish migrants’ previous trauma and its barrier to the potential success of their migration. The interviewees were directly affected by the post-July 15 events in Türkiye. They (39 out of 42) were dismissed from their prestigious positions in governmental jobs in waves of purges after their long service without a logical reason. The rest of the interviewees (3 out of 42) were cadets in closed/transformed schools at that time. Those professionals have been blamed and labeled as “terrorists or sympathizers” by Erdogan’s government. Some of them were detained or jailed following state of emergency decrees, while some others were exiled to hot zones including the Syrian border for unrelated missions with regard to their background and experience. They could not find new jobs after losing their jobs in the country because they were registered in state records. It was a highly difficult decision to migrate from the country, but they understood that it was not possible to have a normal life for themselves or their families in the country. So, even though they faced the risk of death in the migration journey, some of them decided to try. For example,


*I’ve been purged from my job with Erdogan’s law decrees. I stayed two years in jail and was released. After 1-year effort to be able to make work in the homeland, I understood that it was impossible. I got judiciary calls every three months, and the police knocked on my door at inappropriate times. I could not focus on or be motivated to do anything. Employers were reluctant to hire me though I was the best candidate for the positions they were looking for (Male, 40–45, Senior Data Scientist, New York).*


The interviewee has experienced each stage of the difficulties starting with dismissal. He tried to adapt to changes, but it was not possible. In another example, a participant had not experienced detention or jail, but it was still hard to adapt to dramatic changes. He recalled:


*It is like you are a leper. If someone greets you, they can be taken to prison as a “terrorist” [Erdogan’s government labeled them] sympathizer. A few years later, it was the anniversary of 15 July. I didn’t want to go out that day. Lynching culture was very widespread in the country. There were vandals everywhere. But my kid wanted ice cream, so we went out. If they knew that I was a discharged soldier and someone said that they knew me in that crowd and slandered me, they would have killed me next to my wife and child (Male, 30–35, IT professional, Chicago).*


This quote shows the participant’s emotional distress after the events. He was not content with the increasing radicalization and lynching culture at all. He shows the bravery and risk involved in engaging in a simple activity due to his discredited status. Another immigrant was in a professional position to observe the emotional trauma of other fellow citizens. She stated,


*I’ve had my bachelor’s and doctorate degrees at the top two universities in Türkiye. My husband was like me. He was a faculty at a prestigious college. Since we know we are innocent [it was a bad assumption, though], we continued to work at our colleges after 15 July. One week later, my husband was detained for a week [normally, it should be 24 h] in very bad conditions. He doesn’t want to talk about those days. Then, he was discharged from his job, although he was released and acquitted. We opened our own psychology clinic and served people, including those people/families affected by Erdogan’s law decrees. We know they were innocent like us. So, while every other government and private institution rejects serving them, we tried to help them by hiding their identities. I’ve listened to people’s stories, my lips chapped, and I couldn’t get rid of the effects of the people’s stories for days.*



*… after 3 years of judicial control, even though there is nothing new in my husband’s file, the judge decided that he was involved in the incidents [15 July] and gave an arrest warrant for imprisonment. He had to leave without saying goodbye. …I thought if we had a child, he would have a memory left. I thought I couldn’t see him again (Female, 30–35, Psychologist, New York).*


Although professionals, including the participant, are aware of the experiences of other similarly situated individuals, they did not expect the same situations to find them later. Their agency was based on attempting to adapt to the current atmosphere hoping that, in a rational sense, they would be able to be found innocent and continue their careers and way of life. Other interviewees described victimization of psychological and physical torture.


*After 15 July, I worked for a week. Everything was okay. I had a holiday ticket to … [European country] with my wife on 21 July. I asked my supervisor what to do if I should cancel, and he said yes, cancel, wait for the mess. On 22 July, the prosecutor came and took all of us, 400 people got off the buses, and we opened our eyes to Sincan [known jail in Ankara]. There was a right-left police raid like the football players went down to the stadium. I can’t tell you how they cursed, kicked, and slapped. We waited 4 h in the very cold at the tent gym saloon of Sincan. I have seen the tortured state of the friends they took before us… I was sleeping next to the toilet. They gave us a small piece of bread, 1 L of water, and a small, spoiled cheese. So, people needed to go to the bathroom. But they closed most of the restrooms. There were 4 open restrooms for 500–1000 people. We waited in line for 4–5 h. Sometimes the police come and beat someone they choose decently. I didn’t eat for 1.5 days …someone older than me said you couldn’t get anything. We got used to it for 10 days, he said, come and eat mine.*



*…I translated a book [Military Psychology, Second Edition: Clinical and Operational Applications by Carrie H. Kennedy and Eric A. Zillmer] from English into Turkish with my colleagues about the stages of psychological torture when I was working on psychology. They [police] practiced all steps of it. The sub-master relationship has been eroded. They [police] made the soldiers beat the generals. The windows are closed. You don’t know if it’s 8 a.m. or 1 a.m. The time perception has been lost. The spotlights were on, there was a strong sound of air conditioning. Where am I, what am I doing?*



*…They woke me up at night and testify. I said, what expression of what did I do? The lawyers were like prosecutors… The expression has begun. …destroying the constitutional order, bombing the parliament, assassinating the president, deliberately killing people… I said hold on; I was a teacher. Where am I on this story? (Male, 35–40, IT Analyst, New Jersey).*


This statement shows how highly educated professionals were unaware of the events. They were not even aware of their so-called crimes, but they knew what the government was trying to do to professionals. Another participant supports this argument:


*…since we are public lawyers, we are people who have read the history of the coups. How does a coup happen, why does it happen, who does it, and against whom? At that time, there were no conditions for a coup in Turkey. This is exactly what I said to my wife; I said that day [15 July 2016], “If someone is going to stage a coup right now, it will be none other than the President.”*



*…*
*They [police] detained us at the university. They illegally locked us in a room at the university for two days. They mistreated us there. Two days later, at midnight, they took us to a place we did not know before. I can say that it was the day I feared the most in my life. It was a sports hall. The first sight, there was very frightening. Six hundred or seven hundred people. Blood everywhere, blood on the walls, blood on the floors, blood on people’s heads. All dressed in orange. At the bottom, some have black, some brown trousers. Many of them had blood on them. Their jaws, their heads, their eyes… They were all wrapped in bandages. Everyone was still tortured when we left. Screams, crying…*



*… there were two people who were severely tortured in my ward. One of them was the rector [president] of our university. For three months, he could not recover, for three months, he cried. I mean, a sixty years old man, a rector, a professor of medicine was severely tortured… he cried for three months. …he wanted to die. He didn’t think he would ever get out of the prison alive (Male, 45–50, Educator, Florida).*


The experiences they suffered left them little agency except to try to survive. Some focused their worries on the impact on their children, which prompted them to migrate. For example,


*The psychological operation applied to me does not give any results. It can’t make me a terrorist, it can’t radicalize me, but in this environment and this psychology, a child who grows up could be abused by any radical organization in the future. I turned to my wife and said, let’s get out of here. And she said okay (Male 35–40, Security Analyst, Virginia).*


Another statement of an interviewee supports the quote above:


*I was on a short vacation after a work trip in Europe when 15 July happened. My boss called me to return as soon as possible, and I did. However, since I had many courses and missions in the USA and was at the top of my cadre, they’ve looked at me as a potential criminal. The purged officers were, in my profile, successful and hard-working people. Colleagues and superior officers did continuous mobbing on me. The commander of the unit threatened me. Then, the forces command exiled me to a terror region though I completed my mandatory duty in that region. When I went there and see a big poster of Erdogan’s regime related to 15 July, which depicts an unlisted soldier beaten by the so-called public on the Bosphorus Bridge, I asked myself, whom am I working for? For those [public] slaughtering his own Mehmetcik [soldier]. I thought that my children should not grow up in this society (Male, 45–50, Senior Cybersecurity Analyst, Virginia).*


The events following 15 July have changed the feeling of professionals against the government and its supporters. In one case, the interviewee reflected on his extreme situation,


*…one feels trapped. I remember looking at my gun at one point. I wonder if the way to get rid of this pressure is inside the barrel of the gun (Male, 35–40, Data Scientist, Texas).*


In these extreme cases, interviewees could not feel safe. They felt even their human rights are not under protection. For example, one participant was worried about her passport,


*After my husband had to leave home, hide from the police, and prepare to move abroad, police from TEM [anti-terror branch] came to my house at 2 am midnight. We were living in … [high-status people living area]. They [police] went on a terrorist hunt in the middle of the metropolis. It was ridiculous. They wanted to search my home, and I allowed them. I remembered my passport. …how much a passport can be a burden to a person. I wanted to hide it ten floors below the ground. It’s a document that I have a right to take, but, in this atmosphere, they can take it. …and losing it is like losing your future, means losing your spouse, means breaking away from life (Female, 30–35, Psychologist, New York).*


Some interviewees could not easily decide to leave their homeland and thought it might work if they did not see people. They tried to accomplish it by isolating themselves. One recalled,


*…Finally, my friend (ex-helicopter pilot) and I decided to escape from society and raise chickens on a farm far away from the city (Male, 40–45, Senior Data Scientist, Virginia).*


Running away from society did not work for the participants. Most (16 out of 20) exiled immigrants experienced similar cases to those mentioned above statements and decided to risk their lives for the sake of freedom. However, psychological problems and PTSD migrated with some of them, too:


*I see nightmares frequently. Felt like someone would hold my shoulder from the back, and I would experience the same things (Male, 40–45, Pilot, Texas).*


Another participant has seen nightmares, too:


*For a long time, I have seen nightmares. I was in Turkiye in my nightmares. I was angry with myself about why I left my family in the USA and came here [Turkiye] (Male, 45–50, Professor, Texas).*


Since most of the participants had to deal with new problems of the transition to new life, they could not have had a chance to consider the events they experienced until our interview. One reported,


*I spent all my life on duty, honor, and country until my dismissal from service. In order to protect the prestige and interest of the country, we flew many hours over the Aegean Sea to show the Turkish flag and presence, deployed many times to other bases for scramble duties.. During the Syrian crisis, I flew at least 300 hours of air patrolling missions along the border to defend our airspace.. I had to neglect my family and made many sacrifices. We are [crying…] how they can dare to judge our patriotism? How can they dare to label us as “ter…” [terrorists—could not complete the word for 50 s and forced himself not to cry] (Male, 40–45, Business Owner, Texas).*


In another experience, the participant recalled how he tried to deal with his psychological issues by thinking of others who were not as lucky as himself.


*After me, 19 people who tried to cross to Greece from where I passed lost their lives by drowning by boat. Because I had my wife and children with me, I considered myself the happiest and luckiest person in the world. The psychological trauma was a luxury for me, and I needed to relax to live it (Male, 30–35, Senior Cyber Security Analyst, Pennsylvania).*


These psychological traumas migrated with exiled Turkish professionals. They have felt, and some still feel, trauma and developed different strategies to deal with them. As shown in the interviewees’ described experiences, they encountered risk in identification, detention, and other forms of penalties. Many of them discussed either directly facing torture and abuse or the threat of abuse. Their sense of normalcy, safety, and understanding of their human and citizenship rights were drastically taken away. In response, their agency was then refocused on survival. Although migration afforded many a chance at renewed safety, despite the many risks of being detected or the migration journey failing, the trauma was pervasive and impacted their way of life. It also shows that their old way of life and what they were accustomed to culturally, professionally, and socially changed and likely would never be re-experienced. Thus, by examining how trauma impacts the lived realities of migrants, we find that migration may not bring about a full sense of normalcy and sense of satisfaction. Migration may be a new set of insecurities added to the previous insecurities and distress.

### 4.3. Short-Term Poverty

Thus, previous experiences of trauma undermine Turkish migrants’ ability to adjust to a new country and maintain their status. All interviewees experienced financial loss after their dismissal from their previous jobs. However, some experienced short-term poverty fiercely, especially those having an exhausting journey from jail to release and illegal migration to neighboring countries. Some interviewees experienced major financial issues to have this journey. One recalled,


*I had to spend all my money on smugglers and other expenditures to be able to come here (Male, 30–35, Data Scientist, Virginia).*


Another participant had a similar experience to this:


*We were thinking a lot about how to do it [migration] financially. I had a ban on leaving Turkey abroad, and I didn’t have a passport. Even just buying a plane ticket was very expensive. We disposed of our savings, sold the immovables, turned them into cash, and borrowed money from family. I asked for help from friends who went out before me. Someone gave me a phone number [smuggler]. I came to the USA through six countries. When I arrived eight weeks later, all the savings were over, I had $500 in my pocket. I spent $30,000 on flight tickets and smugglers’ money for a passport. We stayed with 6 people in the 1 + 1 house of a friend I know. …after 1 week, I found an hourly job (Male, 30–35, Senior Cyber Security Analyst, Pennsylvania).*


Another participant from a different location experienced similar short-term poverty, though he was on an abroad mission during the events. He recalled,


*Our income sources were completely cut off in an instant. In that context, I started as a taxi driver, a … [one of the top American colleges] graduate taxi driver. I was, at that time, forty-odd years old. But let me say that being able to bring money back to my family, I was lucky enough to experience that after two or three months of interruption (Male 45–50, Professor, Virginia).*


Some other participants experienced short-term poverty and could not buy their essential needs for a while. One stated,


*I had no money to buy furniture for my rented apartment. I was looking for free items on the Facebook marketplace to use or resale (Male, 40–45, Pilot, Texas).*


These four quotes show the general experiences of people who have an exhausting journey from their homeland to the USA. As their experiences show, their financial capital was quickly depleted by the forced migration process. Based on an urgency to flee, less planning could occur, and more risk mounted as they had to find quick ways to escape. A total of 22 out of 42 participants who were abroad during the events still had financial hardships until receiving their work permits and using their savings. Poverty was a completely new phenomenon for all of them, and their once-held high human capital diminished quickly. One stated,


*I came from an upper-middle-class family. I never experienced poverty in my life until my dismissal from the job and losing Türkiye’s sponsorship during my post-doc in the USA (Male, 45–50, Professor, Texas).*


One important reason to overcome short-term poverty as soon as possible was related to the interviewees’ sense of responsibility for their families. Although they had significant enthusiasm and interest to continue their previous careers, they had to be more realistic and make fast economic integration decisions while searching for jobs in the destination areas. One interviewee said,


*If I were alone, I would stay in the park [decrease expenditures to a minimum] and try. However, it was not possible to continue my previous job with my family responsibility and spend more time and money on accreditation (Male, 40–45, Data Scientist, Virginia).*


Another respondent stated,


*If I wanted to continue with piloting, I should freeze the life for 8 months [not earning money in odd jobs], give money on it [credentials in pilot courses], take my time, and make it happen. So, it was a very difficult road for a married man with two children. So, I preferred a cybersecurity job (Male, 30–35, Cybersecurity Analyst, Virginia).*


Here, the desire to pursue his professional work was disrupted by the need to make money and support his family. With depleted resources and an inability to make adequate money, life choices were based on how quickly money could be earned comparable to the previous livelihood. Other interviewees in the data science field have reported reasons similar to the above statement. They could not continue their previous professions while managing the living and household expenditures of their families.

Although they experienced financial issues at this level for the first time in their lives, they were aware that this situation was temporary because their ultimate plan was to rebuild their lives in the U.S. and regain their economic stability and professional status [44]. Additionally, they motivated themselves by thinking about people in their homeland in worse conditions.

### 4.4. Fill the Gap (Overload)

This portion allows us to examine the struggles of the interviewed professionals in regaining their former status and the hardships of finding a professional job. Exiled professionals felt the stress of regaining their previous status and providing ideal living standards for their families. They were overwhelmed and took significant risks for the sake of reaching their goals soon. They understood that they could not do their previous jobs without making many sacrifices and losing time [45,46]. However, some of them (two doctors, two pilots, and six academics) insisted on pursuing their previous jobs by accepting junior-level positions. The rest of them preferred to regain new skills in completely new fields for them, including data science and trade [47]. This process brought extraordinary overload to both groups. One stated,


*I have taught 13 courses at 4 different universities at the same time. I was thinking I needed to earn money; I shouldn’t make my children feel the difficulty of staying here (Male, 45–50, Professor, Texas).*


Another person experienced this overload by doing many jobs. However, this overload also included gaining new skills as fast as possible to reach their former statuses [8,46].


*I did 7–8 different jobs: heavy work in the warehouse, container loading, waiter and cooker in a restaurant, electricity handyman, entertaining children, and delivery. The side works continued for 13 months until November 2020. If there was a scholarship opportunity, it would pay my kitchen expenses and rent for a while, and I could get my license for commercial piloting in 6 months and do my old job. But…it was unrealistic. When I got my work permit, I started working as an Uber driver. It was like a promotion for me. I started a fast boot camp in the cyber security field. I’ve been working hard for 6 months. I was able to provide for my family’s livelihood by doing Uber for 4–5 h a day. I attended most of the classes with my headset while driving an Uber. 2 weeks after the course ended, I found a six-figure work (Male 35–40, IT Security Specialist, Virginia).*


Our participants shared similar experiences about doing daily jobs to make money as much as they could and making sacrifices to learn new jobs [22,45]. The following statement shows the general behaviors of the interviewees:


*While driving and waiting for my working permit, I was taking data science camp courses at night and watching as many tutorials and courses as I could to learn more about the field. I did not know even basic terms in the field like “terminal” I googled everything I heard new. Listened to very short free learning videos at first and watched longer ones when I found a chance. After a year-long effort, I was familiar with almost everything, and people were asking me about any issues they faced (Male, 40–45, Senior Data Scientist, New York).*


Although it was hard to transition their previous skills into new jobs directly, their backgrounds, hardworking skills, and other soft skills were rewarded in their new jobs. For example, the following statements support this transition process:


*Because my previous experiences [20 years of engineering in government] are different from the field I applied for, I started in a junior position. However, I moved to senior positions faster than the others. You can reflect on your experience after entering the first job (Male, 40–45, Senior Software Engineer & Team Leader, Virginia).*



*I made the delivery half day to make money and followed Udemy, Coursera courses, and YouTube videos in data science for a year. Applied for any job post for three months and took an offer from three different companies and started to one of them and changed it with a better job option in another company. Since I’ve felt myself behind in the new field, I forced myself to learn all new terms I’ve heard daily. When our leading senior data scientist with ten-plus years of experience left for another job, the manager asked me to take over his role (Male, 40–45, Senior Data Scientist, New York).*


The participants were rewarded for their hard work after finding their first job or starting a new position in a new field in a short amount of time. Some interviewees even took significant financial risks to regain their former statuses in a short amount of time. For instance,


*I took over a store with a debt of $250 k. I did two-man jobs for three years; I had to learn design, sales and how to run a business; we closed the debts, and we made a profit. We are doing kitchen and bathroom design very well (Male, 40–45, Business Owner, Texas).*


His previous experiences were completely different from the requirements of the new job. However, he was aware of his soft skills to learn new things quickly. Therefore, this risk was reasonable for the participant. However, some interviewees could not accept this career change easily:


*When I came here, I thought that burning the diplomas they were useless. … whomever I talk to says the same thing. My left brain is convinced to pursue the IT course, but my right brain says no, you have worked so much for psychology, you have spent years, and you should continue working on it. (Female, 30–35, Psychologist, New York).*


The participants and other highly educated immigrants know the market needs in IT fields in the USA. Their background, math skills, and education are advantages in finding a job in these fields. However, it is not an easy decision, especially for academics. On the other side of the same coin, we have found the negative consequences of this filling-the-gap process. This kind of overloading and risk taking caused a loss of well-being for exiled immigrants. Some reported,


*I gained a lot of weight in these two years. I have been working hard with an ambition to close the gap. This process left permanent effects on my health.*



*COVID-19 made this process harder for professionals.*



*In times of lockdown due to COVID, I was going for handyman jobs like the assembly of trampolines or basketball hoops and risking my life and health to make money for my family (Male, 40–45, Senior Data Scientist, New York).*


Another statement from a different location reflects a similar feeling:


*While everyone sitting in their homes and trying to avoid COVID, I was making food delivery in Instacart (Male 35–40, IT Security Specialist, Virginia).*


Moreover, after finding an ideal full-time job, interviewees (33 out of 42) encouraged their spouses to find a full-time job, especially in the data science field (e.g., cybersecurity, QA, Salesforce), by taking more responsibility at home. Then, many spouses (21 out of 33) found full-time jobs in these fields. Some spouses preferred to pursue their education in new or previous fields, especially in the fields of law and education, while some of them transitioned their hobbies to online store jobs. One interviewee observed this change in spouses,


*90% of my [exiled] friends’ spouses found full-time jobs in our region [New York] (Male, 40–45, Senior Data Scientist, New York).*


Most of the interviewees reported that they continued their side jobs and extra work even after getting very good jobs. The motivation that lies behind this may be related to their connection with people in their home country. Interviewees’ motivation to help others and support professionals in similar situations encouraged them to take more risks and start new initiatives. These initiatives supported silenced and isolated people emotionally and financially in their homeland and abroad [48]. For example, the following statement shows the psychological support they provide for others:


*We’ve founded a psychological institute with volunteer psychologists and doctors to help forced migrants and people who experienced trauma. We’ve provided free service to 1400 people so far. After two years, we also started to help people who were forced to migrate all around the world (Female, 30–35, Psychologist, New York).*


Another statement shows the support for gaining new skills and financial well-being. The participant recalled,


*You know professionals like us [discharged with Erdogan’s law decrees] cannot do any work in Turkiye. So, while gaining new skills in a new country, I’ve also thought of giving away to people in Turkiye and here [USA]. I founded an IT school with my friends, we worked very hard and slept less, and within four years, we were selected as one of the top IT schools in the U.S. and Europe (Male, 40–45, Founder of an IT school, Virginia).*


The experiences of the participants were similar to each other in terms of regaining their statuses. They worked hard and used every opportunity to accomplish their goals as soon as possible. Although they encountered challenges finding temporary income and regaining their professional credentials or seeking new education and credentials altogether [44], one aspect that does stand out is their agency evident in their ability to take on innovative strategies, commit to initiative and motivation, and find success. This ability is linked to their professional credentials and experiences, and the ability for them to build on previous social and cultural capital to be successful. However, despite their success, there is concern that their profession and educational skills were not automatically deemed verifiable and usable, and the time lost either regaining credentials or seeking new careers reveals a loss of earning potential, lower job and life satisfaction, a poorer quality of life generally, and a drop in occupational commitment [45,46].

## 5. Conclusions

The findings of this study highlight that Turkish migrants experience challenging conditions throughout the migration period, beginning with the violence that prompts them to migrate quickly out of safety concerns, the short-term poverty they experience in the migration journey itself and in settling into the host country, and the temporary loss of professional jobs and status as they engage in the credentialing process. We anticipated both the challenges that come with urgent forced migration, as has been well-documented in the literature regarding the dire plights immigrants face when forced to flee, and the economic assimilation liminality that comes with finding new jobs and adjusting to a new society [8,16]. However, the findings give us a more nuanced understanding of the insecurities and barriers within the migration process. First, higher forms of human capital are not as stable; however, high human capital can increase agency. Despite having economic security, specifically financial capital, namely income, savings, and resources, Turkish immigrants found that this was depleted quickly at the start of the migration process. Their ability to receive income to support their entry and settlement experiences in the U.S. was not a stable resource. However, due to high human capital, they could rely on professional skills, experience, and networks to regain their economic security and career, leading to an opportunity to regain their quality of life economically [8,44]. Second, the insecurities in each stage of the migration process (deciding to migrate, migration journey, and assimilation/settlement) are compounded because they build on each other. Thus, the process is not made of discrete issues that can be resolved before addressing any new insecurities. In the case of many Turkish migrants in this study, trauma was an ongoing component in their life, and the economic and professional loss that occurred, although mostly temporary, ended up having long-term consequences: the stress of rebuilding, the loss of their professional status, and the lack of presence with their family, all of which impacted their quality of life. Economically, it meant the loss of earnings, and emotionally, the inability to engage in mental health intervention to improve emotional well-being. One last conclusion we found relates to their new way of life in relation to their previous sense of identity. In addition to many Turkish migrants seeking new fields for their careers, all experienced a change in their way of life. Their upbringing, connection to their homeland, friends, family, acquaintances still residing in Türkiye, their cultural customs, and way of life can never be returned to in the same way they had once experienced them. The harmful and lasting effect of events, such as the coup, has reshaped their cultural connection, pride, and well-being associated with their homeland. Even though we could generally imply that the U.S. and other European countries promise safety from the regime in Türkiye, their trauma and insecurities are replaced with new insecurities as they adjust to a different culture.

This study provides a greater understanding of immigrants’ agency to regain professional status, mostly using their own professional skill sets, experiences, and social networks. This study provided a broader context to exploring the trajectory of the migration process as immigrants sought to regain their professional livelihoods in the U.S. The study also is comparable to other studies that focus on specific Western countries, particularly Canada and Western Europe, by showing both the challenges in transferring skills and education and the short-term disruption to regaining their professional careers.

In addition to the important contributions of the paper, it has some limitations in the generalizability of the results. The snowball sample includes many professional participants representing Türkiye in Western countries. Therefore, they may have better language, education, and professional skills than other highly educated people in the general migration population after the 15 July events. Interested researchers can test our findings that migration can bring new insecurities to previous issues starting with pre-migration decisions of highly educated immigrants. Quantitative and qualitative inquiries on the integration process and degree of agency can help provide an understanding of the short- and long-term impacts of deskilling, seeking new education and credentials or transferability of credentials, and the earnings gap. Further, highly educated immigrants from economically disadvantaged countries may deplete their financial capital to reach the U.S. through illegal ways, but they may depend on their high human capital and network for economic integration since the U.S. provides many opportunities to use them accordingly. Both comparative studies across countries and country-specific economic integration can be further extended to build on findings in the literature that suggest integration is underdetermined by a multitude of economic and social factors, policies, immigrant reception, and types of professions. Future research may test these findings in other highly educated migrant groups such as the recent highly educated Ukrainian population who were forced to migrate to the U.S. and other European countries. To further understand the process of exiled professionals, future research could study their legal status experiences as well as the comparison between their assumptions and facts since their migration decisions were unprepared and quick. The nuanced understanding of the experiences of selected exiled professionals provided us the opportunity to view the policy issues they faced. These professionals offer the U.S. a unique labor stock in less-found expertise areas. However, policy reform is needed to facilitate the immigration status of migrants and create a smooth integration process by lessening the time taken to verify foreign credentials and enabling professionals to continue in their line of training and career.

## Data Availability

Not applicable.

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
