# Peer review of "Reskilled and Integrated, but How? Navigating Trauma and Temporary Hardships"

_ijerph, 2022, doi:10.3390/ijerph192013675_

Round 1
Reviewer 1 Report
This is a useful paper focusing on “the economic assimilation experiences of forced yet highly educated Turkish immigrants [N=42] to better understand complex forces that either aid or hinder economic integration in a destination society [U.S.A.].” This paper expands the existing “forced migration” literature in North America and it fill gaps in the literature. It also has policy implications. As the author(s) note: “the process [forced aspects of migration] has positive and negative components, but understanding this nuanced process provides opportunities for policy reforms that can shorten the time of re-education, increase employability, and support well-beings.” (p. 1). Its research methods (e.g., face-to-face interviews with 42 highly educated Turkish people in four U.S.A. cities; use of snowball sampling techniques…) are adequate; and its conclusions/findings are relevant for both academics, governments and policymakers. My overall assessment is that this paper would make an important contribution to multidisciplinary scholarship on ethnic and migration studies, as well as an informative reading for the audience of the International Journal of Environmental Research and Public Health, but only if some major revisions are undertaken by the author(s).
Specific Comments:
a) Introduction (pp. 1-3). There is room here to expand on why this research topic is important and why the focus is in four U.S.A. cities (New Jersey, Houston, Dallas and Virginia Beach). What makes these cities so unique? Provide more socio-demographic information on Turkish migrants in these cities, including more information on their settlement experiences (e.g., including major barriers/challenges encountered on arrival...). What are the most important research questions guiding this study? In sum, there is room for a stronger rationale/justification of the research topic, study population and the study areas.
b) Literature Review (pp. 3-5): The author(s) show familiarity with the literature. However, I recommend they expand the literature review by focusing more on the barriers/challenges and coping strategies faced by forced migrants in U.S.A. Comment on the main “push-pull” forces that may contribute to the settlement of forced migrants, including highly educated Turkish people in major U.S.A. urban centres and the numerous challenges they may face in integrating into the host society. Forces at play? Identify gaps in the literature and expand on how this study fill gaps in the literature.
c) Materials and Methods (pp. 5-7): The author(s) made good use of the sources used (e.g., face to face interviews...). However, there is room to expand on (a) the first study conducted in 2019. How many Turkish migrants participated in this study. Specify the study areas and sampling strategies, etc., (b) with regard to the present study - why the choices of New Jersey, Houston, Dallas and Virginia Beach as the study areas? Add socio-demographic information about the Turkish population in these study areas, and c) Why the inclusion of 12 interviewees from previous comparative study (2019)? Specify cities where this study took place. Also justify rationale for selecting these 12 Turkish migrants who participated in a 2019 study.
d) Results and Discussion (pp. 7-15). Descriptive...but very interesting results/findings. I enjoyed reading this section of the paper. However, the sub-section “Short-term Poverty” (pp. 11-12) is too short! Less than one page. What can we learn from migrants’ experiences/poverty? Expand. Question - to what extend the results of this study corroborate or not other North American studies?
e) Conclusion (p. 15): The author(s) can improve this part of the paper by expanding on how this study fill gaps in the North American literature, limitations of this study (e.g., small sample, etc...) and on “areas for further research.”
Reviewer 2 Report
Thank you for the opportunity to revise this paper titled: “Reskilled and Integrated, But How? Navigating Trauma and Temporary Hardships”. I think the authors are addressing a very interesting topic and the paper seems overall well conducted.
As some comments that I hope the authors take as constructive, I have the following:
The coup in Turkiye plays an important role in the research conducted as well as in the findings. However, it is not mentioned in the abstract, perhaps it should be?
In terms of the population of the sample, to what extent the fact they were all married may affect the results? Also, were they married to other Turkish, American, or other nationalities? And how that may matter?
Typically, qualitative research intends to open up avenues for further research and develop established theories. How do your results bring opportunities to interested researchers to test, expand, refine, etc existing streams of literature?
Good luck with your research!
Reviewer 3 Report
The article presented is relevant and necessary.
1º The review of the literature presented in the article is very scarce considering the number of investigations and proposals for systematic reviews carried out in the international field on the subject. For future research, a comparative analysis with other continents and the determining factors could be carried out. In any case, the revised bibliography is extensive but the lines of work are not reflected in a structured way.
2º The research design is appropriate. It would be interesting to expand the information by designing studies where the research was of a quantitative or mixed nature, as a result of the results presented in the research. The coding system can be improved. It is also necessary to use Nvivo or AQUAD qualitative analysis software or any other that the researchers deem appropriate and that offers us clarity in the analysis carried out.
Round 2
Reviewer 1 Report
--
Reviewer 2 Report
I have no further comments